# Advances in Nucleic Acid Research: Exploring the Potential of Oligonucleotides for Therapeutic Applications and Biological Studies

**DOI:** 10.3390/ijms25010146

**Published:** 2023-12-21

**Authors:** Maria Moccia, Barbara Pascucci, Michele Saviano, Maria Teresa Cerasa, Michael A. Terzidis, Chryssostomos Chatgilialoglu, Annalisa Masi

**Affiliations:** 1Istituto di Cristallografia, Consiglio Nazionale delle Ricerche, Strada Provinciale 35d, n. 9, 00010 Montelibretti, Italy; maria.moccia@cnr.it (M.M.); barbara.pascucci@cnr.it (B.P.); 2Istituto di Cristallografia, Consiglio Nazionale delle Ricerche, URT Caserta, Via Vivaldi 43, 81100 Caserta, Italy; michele.saviano@cnr.it; 3Istituto di Cristallografia, Consiglio Nazionale delle Ricerche, Via Giovanni Amendola 122/O, 70126 Bari, Italy; mariateresa.cerasa@cnr.it; 4Laboratory of Chemical Biology, Department of Nutritional Sciences and Dietetics, Sindos Campus, International Hellenic University, 57400 Thessaloniki, Greece; mterzidis@ihu.gr; 5Istituto per la Sintesi Organica e la Fotoreattività, Consiglio Nazionale delle Ricerche, 40129 Bologna, Italy; chrys@isof.cnr.it; 6Center of Advanced Technologies, Adam Mickiewicz University, 61-712 Poznań, Poland

**Keywords:** nucleic acid, DNA, microRNA (miRNA), small interfering RNA (siRNA), peptide nucleic acid (PNA), antisense oligonucleotides (ASOs), 5′,8-cyclopurines (cPu), ODNs

## Abstract

In recent years, nucleic acids have emerged as powerful biomaterials, revolutionizing the field of biomedicine. This review explores the multifaceted applications of nucleic acids, focusing on their pivotal role in various biomedical applications. Nucleic acids, including deoxyribonucleic acid (DNA) and ribonucleic acid (RNA), possess unique properties such as molecular recognition ability, programmability, and ease of synthesis, making them versatile tools in biosensing and for gene regulation, drug delivery, and targeted therapy. Their compatibility with chemical modifications enhances their binding affinity and resistance to degradation, elevating their effectiveness in targeted applications. Additionally, nucleic acids have found utility as self-assembling building blocks, leading to the creation of nanostructures whose high order underpins their enhanced biological stability and affects the cellular uptake efficiency. Furthermore, this review delves into the significant role of oligonucleotides (ODNs) as indispensable tools for biological studies and biomarker discovery. ODNs, short sequences of nucleic acids, have been instrumental in unraveling complex biological mechanisms. They serve as probes for studying gene expression, protein interactions, and cellular pathways, providing invaluable insights into fundamental biological processes. By examining the synergistic interplay between nucleic acids as powerful biomaterials and ODNs as indispensable tools for biological studies and biomarkers, this review highlights the transformative impact of these molecules on biomedical research. Their versatile applications not only deepen our understanding of biological systems but also are the driving force for innovation in diagnostics and therapeutics, ultimately advancing the field of biomedicine.

## 1. Introduction

Nucleic acids are biologically significant molecules that play an important role in the storage, transmission, and expression of genetic information within living organisms. These molecules, with DNA and RNA as maybe the most recognizable, are essential for various biological processes and are often referred to as the building blocks of life [1]. They are composed of nucleotide units, each comprising three essential components: a phosphate group, deoxyribose or ribose (DNA or RNA), and a nitrogen-containing base (adenine (A), thymine (T), guanine (G), and cytosine (C) in DNA, while in RNA, uracil (U) replaces thymine).

The linear arrangement of these nucleotides forms the backbone of nucleic acid chains, with the nitrogenous bases serving as the information-carrying entities, coding the genetic instructions essential for cellular functions, including protein synthesis and heredity. Understanding the composition of nucleic acids is pivotal in elucidating their roles and significance in the realm of molecular biology and genetics.

In recent years nucleic acids have garnered significant attention as versatile biomaterials with immense potential in diverse biomedical applications. Their unique properties and functional capabilities make them invaluable in fields such as gene regulation, biosensing, drug delivery, and therapy [2]. ODNs are defined as polymers of nucleic acids composed of a relatively small number of nucleotide units, typically fewer than 100. Thanks to their versatility and programmability, ODNs have emerged as powerful tools in various fields, including therapeutics, diagnostics, and biotechnology. In the field of drug development and precision medicine, these short nucleic acid sequences, modified or unmodified depending on the intended use and source, hold immense potential since they can specifically target and modulate gene expression [3,4,5]. This class of substances utilizes a range of mechanisms to address various medical challenges, offering a diverse toolkit that includes gene therapy (siRNA, ASOs, plasmid DNA, miRNA, shRNA: short hairpin RNA, mRNA: messenger RNA, and CRISPR/Cas9), RNA-based vaccines, and DNA vaccines (third-generation vaccines) [6,7]. In particular, siRNAs and ASOs offer precise control over gene expression. By targeting specific mRNA sequences, they can inhibit the translation of target proteins, allowing for the regulation of disease-associated genes, with profound implications for the treatment of genetic disorders and various diseases, including cancer [8,9]. RNA-based vaccines (pioneered by Katalin Karikó and Drew Weissman, who were awarded the 2023 Nobel Prize in Medicine), as exemplified by the COVID-19 mRNA vaccines, represent a revolutionary approach to vaccination, harnessing nucleic acids to trigger immune responses. This alteration has the potential to inhibit the activation of inflammatory reactions and enhance protein production upon mRNA delivery to cells [10,11]. In biosensing, ODNs have been harnessed as molecular recognition elements. DNA and RNA aptamers, for instance, exhibit high specificity and affinity for their target molecules, enabling the sensitive detection of biomarkers and pathogens. This has revolutionized diagnostic tools, offering rapid and accurate disease detection [12,13,14]. These innovative approaches hold great promise for addressing an array of genetic diseases and infectious threats. 

Additionally, nucleic acids are used as vehicles for drug delivery, as they facilitate the transport of therapeutic agents to specific cells or tissues. This targeted drug administration minimizes the undesired off-target effects and enhances treatment efficacy [15]. Stability, target specificity, cellular uptake efficiency, immunogenicity, off-target effects, and overall pharmacokinetics represent crucial parameters that need to be considered when designing oligonucleotides for specific applications [2]. Indeed, although nucleic acids exhibit significant potential for both therapeutic and diagnostic applications, their vulnerability to degradation by nucleases and their physicochemical characteristics pose challenges that impede the effective delivery of nucleic acids into cells for therapeutic purposes. To address these limitations, new technologies have emerged such as the development of DNA-based nanocarriers and lipid-based formulations which can encapsulate and safeguard nucleic acids from degradation, thus facilitating their release into cells [16].

Additionally, scientific evidence demonstrates that bioconjugation, involving proteins, peptides, antibodies, small molecules, nanoparticles (NPs), or other biological molecules, enhances the delivery of ODNs [17]. To improve their stability, binding affinity, and half-lives, over the previous decades, various oligonucleotides have been synthesized through chemical modifications to their phosphodiester backbone, sugar moiety, nitrogenous base, or in combinations. 

The most common chemical modifications of DNA/RNA backbone currently in use are the 2′-O-methyl (2′-OMe), 2′-O-methoxyethyl (2′-MOE), 2′-fluoro (2′-F) (Figure 1). These modified forms exhibit higher stability compared to unmodified RNAs [18]. Locked nucleic acids (LNAs) were also developed to enhance the efficiency of the synthetic ODN to target miRNAs [19,20]. Some recent reviews reported the use of chemically modified sequences for targeting miRNAs [21,22] and described the primary challenges associated with the delivery of RNAi drugs [23]. Another nucleic acid analogue is PNA, in which the sugar phosphate backbone is replaced by an achiral, neutral N-(2-aminoethyl) glycine, and the nucleobases are linked via a methylene carbonyl linker to the main backbone (Figure 1) [24]. 

The chemical structure of PNA provides the same distance between nucleobases, allowing the efficient hybridization with complementary sequences of both DNA and RNA. Due to their neutral backbone, PNA sequences exhibit higher binding affinity and better stability with the complementary DNA/RNA. This unique property enables PNAs to strongly bind with their RNA or DNA targets, using antisense and antigene approaches to effectively regulate gene expression [25]. In the context of the antigene mechanism, PNA can interrupt transcriptional processes by forming various complexes with double-strand DNA (dsDNA). These complexes include a stable triplex structure, strand invasion complexes, or strand displacement complexes that cause hindrance to the RNA polymerase structure. Triplex-forming PNA may prevent duplex DNA unwinding or inhibit the binding of a transcription factor to the promoter region of specifically targeted genes, which can affect transcription. Additionally, blockage of RNA polymerase elongation can occur by invasion structures within the transcribed region on the template strand [26]. In the antisense approach, PNA molecules employ steric blocking mechanisms to exert inhibitory effects on the translational process. The therapeutic applications of this mechanism will be discussed in detail in Section 2.2 of this review. Another distinctive feature of PNAs is their resistance to the hydrolytic activity of nucleases and proteases [27]. For all these reasons, since their discovery, PNAs have been used for therapeutic [28] and diagnostic [29] purposes and more recently as anti-miRNA-based therapeutics [21,30] and target protectors [22,31]. 

Other notable nucleic acid analogues include morpholino phosphorodiamidate oligomers (PMOs) [18,19] and unlocked nucleic acids (UNAs) [30] (Figure 1). PMOs replace the deoxyribose sugar with a morpholine ring and utilize phosphorodiamidate linkages, offering enhanced stability and resistance to nucleases. UNAs, on the other hand, introduce a flexible methylene linker between the 2′ and 4′ positions of the ribose, providing an adjustable level of duplex destabilization. Both PMOs and UNAs contribute to the expanding repertoire of chemical modifications, offering tailored options for optimizing the performance of synthetic oligonucleotides in diverse applications. 

This review aims to summarize recent advances in the biomedical applications of oligonucleotides, encompassing both native nucleic acids and chemically modified nucleic acid analogues. Specifically, recent developments in the use of ASOs, miRNAs, siRNAs, LNAs, and PNAs are reported. Additionally, we highlight the role of oligonucleotides as biomimetic models for identifying novel biomarkers of radical damage.

In fact, oligonucleotides are also widely used for in vitro studies due to their ease of synthesis and their ability to mimic the structure and properties of native DNA and RNA molecules. They constitute a valid tool for investigating interactions with biomolecules, including proteins and enzymes, as well as for studying the effects of mutations or damage on the structure of DNA or RNA and their consequent impact on functionality and efficiency [32,33]. Herein, in addition to the use of oligonucleotides in the therapeutic field, we will focus on exploring the application of oligonucleotides as biomimetic models for the study of the cPu and their repair. cPu lesions exert profound effects on genomic stability and mutagenesis and contribute to the pathogenesis of various diseases. Extensive research has revealed their association with a wide range of disorders, including cancer, neurodegenerative diseases, and aging-related conditions [34,35]. Elucidating the specific mechanisms through which cPu lesions influence disease development is crucial for developing targeted therapies and preventive measures. Hence, in addition to summarizing recent advances in the biomedical applications of oligonucleotides, we present recent studies and findings related to this lesion and examine the applications of the biomimetic oligonucleotide model in identifying new biomarkers and conducting repair studies [33,36,37,38,39,40].

## 2. Oligonucleotides as Therapeutic Agents

### 2.1. Therapeutic Applications of Antisense Oligonucleotides 

ASOs are synthetic ODNs, usually 8–30 nucleotides long, that are specifically designed to bind to premessenger RNA (pre-mRNA) and/or mRNA by Watson–Crick base pairing, resulting in a high specificity to the target. They represent potential therapeutic agents for various human diseases. ASOs, unlike traditional drugs, interact with pre-mRNA and/or mRNA and modulate protein expression, offering the potential to halt protein malfunction and eliminate the source of disease. In the last decades, considerable efforts have been made to find new ASOs with a maximum therapeutic profile to treat rare and genetic disorders, cancer, viral infections, and neurodegenerative disorders (Figure 2) [41,42,43].

Although the design and preparation of ASOs involve several steps, they can be synthesized quite quickly, generally using solid-phase methods. First, the specific mRNA target or gene sequence to be modulated for therapeutic purposes must be identified, then the oligonucleotide sequence (DNA or RNA) complementary to the target mRNA must be designed and synthesized. To improve ASO stability, binding affinity, and resistance to degradation by nucleases, chemical modifications can be introduced. Common modifications include phosphorothioate backbone linkages and nucleotide modifications such as LNA, PNA, and PMO [44,45,46]. Comprehensive studies revealed that ASOs can exert their effects through two distinct mechanisms: RNA cleavage and RNA blockage.

In the RNA cleavage mechanism, ASOs are specifically designed, targeting an RNA molecule through complementary base pairing. Once bound, ASOs recruit cellular enzymes called ribonucleases, which cleave or cut the RNA molecule at the ASO’s binding site. This cleavage leads to the degradation of the RNA, preventing its translation into protein. This mechanism is particularly effective for targeting disease-causing RNA molecules, such as those associated with genetic disorders or viral infections.

ASOs can also work by blocking the normal function of RNA molecules without causing their cleavage (RNA blockage mechanism). In this mechanism, ASOs bind to the target RNA molecule, typically at specific regions such as the untranslated regions (UTRs) or regulatory sequences. By occupying these binding sites, ASOs interfere with the normal interactions between the RNA and cellular factors, such as ribosomes or other proteins, which are essential for their proper function. This disruption can prevent the RNA from being translated into protein or alter its processing, stability, or localization. RNA blockage is often employed to modulate gene expression, regulate protein production, or correct aberrant splicing events. The choice of mechanism depends on the specific therapeutic goals and the characteristics of the target RNA molecule [42,47].

Research is ongoing to explore the selective action of ASOs, which results in fewer side effects and less toxicity than traditional drugs [41]. ASOs encounter significant challenges during application due to their limited ability to permeate the plasma membrane. To be effective, they must possess some fundamental characteristics, including resistance to degradation, avoidance of renal clearance, escape from protein sequestration, ability to cross the capillary endothelium and the intracellular plasma membrane, resistance to degradation in the lysosome, and the capability to cross the blood–brain barrier (BBB) for central nervous system (CNS) treatments [4]. Until now, numerous ASO-based therapies have focused on localized delivery, for example, targeting the eye or spinal cord, where it is easier to reach the target. The eye, considered an immune-privileged organ, has been extensively studied for ASO therapies (e.g., pegaptanib and fomivirsen). In contrast, for central nervous system applications, the effectiveness of direct injections of ASOs into the cerebrospinal fluid via lumbar puncture (e.g., nusinersen) has been shown [48]. Furthermore, the liver, with its robust perfusion and presence of receptors that facilitate absorption, serves as a prevalent release site for ASOs. As described above, developing safe and selective methods to cross the BBB represents a significant challenge. Moreover, it is necessary to ensure that ASOs specifically target RNA sequences associated with neurodegenerative diseases to avoid off-target effects that might lead to undesirable outcomes. Additionally, the development of effective delivery systems capable of targeting specific brain regions and delivering ASOs in therapeutically relevant quantities represents an additional challenge. Minimizing the immune responses triggered by ASOs is also crucial to preventing adverse reactions in patients [49]. To overcome these challenges, new studies are exploring a range of methods. Advances in sequencing technologies and computational biology enable more precise target identification and design. Using nanoparticles, viral vectors, or exosomes as delivery vehicles can enhance the targeted delivery of ASOs to specific brain regions, improving cellular uptake and reducing degradation [50,51]. Chemical modifications can also reduce immunogenicity, while rigorous testing and modifications are essential to create ASOs with minimal immune response [52,53]. Several ASOs have been introduced targeting neurodegenerative and neuromuscular disorders, although the development of ASOs for the latter diseases is a complex and ongoing process, involving interdisciplinary collaboration, extensive testing, and regulatory oversight [54,55]. 

In the context of Huntington’s disease (HD), antisense therapies aim to lower the expression of the Huntingtin protein. The innovative approach involves single-stranded ASOs utilizing the RNase H1 mechanism [56,57]. Currently, three drugs are undergoing clinical trials: NCT02519036, NCT03225846, and NCT03225833 [55,58]. Regarding Alzheimer’s disease, therapies aimed at disaggregating and eliminating β-amyloid have been a focal point of pharmaceutical investigation over the last two decades, albeit with restricted success [59]. The strategy of ASOs is to decrease the intracellular load of tau, thereby decreasing the creation and dissemination of intracellular neurofibrillary tangles [60]. Comparable to observations made with huntingtin-targeted ASOs, a 2′-MOE-modified gapmer ASO against tau RNA demonstrated a dose-dependent decrease in tau expression, with enduring effects for more than four months. PS19 transgenic mice carrying the human P301S mutation were found to have tau aggregates prevented and reversed after administration of tau-targeted RNase H ASO. Following toxicological studies, a phase 1 clinical trial was initiated for a microtubule-associated protein tau-targeted ASO in patients with mild Alzheimer’s disease (NCT03186989) [55]. 

Over the past twenty years, the progress in the development of therapies for amyotrophic lateral sclerosis (ALS) has been rapid. Hereditary types of ALS represent an interesting prospect for antisense therapies, as targeting the mutant protein or RNA is a clear objective with minimized risk.

Currently, clinical trials are underway for ASOs targeting SOD1, C9orf72, FUS, and ATXN2, which are associated with familial or sporadic forms of ALS. ASOs using RNase H-dependent cleavage mechanisms are predominantly employed in preclinical testing and clinical trials for ALS [53,55]. Another important application field in which ASOs have been studied is the potential treatment of cancer. Their main mechanism of action is based on binding to specific genes that are overexpressed or mutated in cancer cells, leading to tumor growth suppression and cell death. An example of this is the use of ASOs to target the oncogene *B-cell lymphoma 2 (Bcl2)*, which has been observed to stimulate apoptosis in cancer cells and inhibit tumor formation.

ASOs have also been used to target miRNAs that control gene expression and are linked to cancer. A preliminary investigation successfully suppressed mouse lung cancers by delivering tumor suppressor miRNA mimics using ASOs incorporated in a neutral lipid emulsion [61]. ASOs have been widely employed in nonsmall cell lung cancer (NSCLC) therapy to target key gene regulators involved in the malignant phenotype. Some of the targeted genes include *Bcl2*, *protein kinase B*, *Kirsten rat sarcoma virus* (*KRAS*), *vascular endothelial growth factor*, *signal transducer and activator of transcription 3*, *clusterin*, and *protein kinase C alpha* (*PKC alpha*) [62]. Currently, aprinocarsen (NCT00017407, NCT0034268) and custirsen (NCT01630733, NCT00138658) are under clinical trials targeting *PKC alpha* and *clusterin*, respectively, which are associated with NSCLC.

Inflammatory diseases encompass a broad range of conditions characterized by chronic inflammation and immune dysregulation. ASOs have emerged as a promising therapeutic strategy for addressing these conditions. Rheumatoid arthritis, inflammatory bowel disease (including Crohn’s disease and ulcerative colitis), and psoriasis are examples of inflammatory diseases where ASOs have demonstrated potential benefits. They can be designed to target specific genes or gene products involved in the inflammatory response, thereby modulating the immune system and reducing inflammation. ASOs targeting specific cytokines, such as tumor necrosis factor-alpha, interleukins (ILs), and interferons, have been investigated for conditions like rheumatoid arthritis, Crohn’s disease, and psoriasis. By blocking the production of these cytokines or their receptors, ASOs can dampen the pro-inflammatory signaling cascades that drive disease progression [63]. In particular, Karras et al. investigated the anti-inflammatory effect of an inhaled IL-4 receptor-α antisense oligonucleotide (IL-4Rα) in mouse models. By reducing IL-4Rα protein expression in various lung cells, including eosinophils, dendritic cells, macrophages, and airway epithelium, the modified IL-4Rα showed promise in mitigating allergic lung inflammation and airway hyperreactivity associated with asthma/allergy [64]. The investigation of *SMAD family member 7* (*Smad7*) ASOs as a potential treatment for inflammatory bowel disease by inhibition of transforming growth factor-β1, which plays a role in mucosal inflammation in the gut, *Smad7* ASOs were found safe and tolerable by patients with Crohn’s disease [65]. 

Additionally, ASOs can be used to modulate the activity of chemokines, which are involved in the activation and recruitment of immune cells during inflammation. A particular type of ASO called 2′-deoxy-2′-fluoro-D-arabinonucleic acid ASO was studied in a mouse model of spinal cord injury (SCI) to inhibit a specific chemokine called CCL3. This was shown to reduce inflammation at the site of injury, paving the way for possible therapies to treat SCI and other central nervous system disorders. Additional investigations are required to comprehensively assess the therapeutic potential and safety of this approach in the clinical setting [66].

### 2.2. Therapeutic Applications of RNA Interference (RNAi)

RNAi is a process, which uses double-stranded RNA in a sequence-specific way, that implies the mRNA degradation and the suppression of target gene expression. RNAi can be achieved by either siRNA or miRNA. 

The miRNA and siRNA are small molecules of RNA, with the former ranging in length from 19 to 23 nucleotides and the latter from 20 to 25 nucleotides, that are implicated in RNA interference and work through similar mechanisms: they are processed by Dicer inside the cell and are incorporated into the complex RNA-induced silencing complex (RISC) (Figure 3) [67,68]. siRNAs, called short interfering RNA, are a class of double-stranded noncoding RNA molecules, and they have primary roles in biology [69]; miRNAs are single-stranded, noncoding molecules from endogenous noncoding RNA, meaning that they are synthesized inside the cell. Some differences between these two molecules can be spotted: siRNAs block the expression of one specific target mRNA, whereas miRNAs regulate the post-transcriptional expression of multiple mRNAs depending on their mechanism of imperfect complementarity, except the seed region (which is located at 5′ miRNA), which must be perfectly complementary. Another important region of miRNA is called 3′ supplementary, which encompasses nucleotides from 12 to 16; this region is important for target recognition [70]. 

siRNA therapeutics are based on the introduction of synthetic double-stranded RNA, which cause the inhibition of a specific gene mRNA to generate gene silencing. The design of siRNA is quite difficult; the gene silencing caused by siRNA depends on the region of mRNA that is complementary. Understanding the relationship between siRNA and mRNA binding can facilitate the design of siRNA with optimal efficiency. Many siRNA design algorithms have been developed to predict the efficacy [71,72]. However, the efficacy must be validated by experimental methods. One of the major drawbacks of using siRNA can be the off-target effects due to downregulation of unintended, unpredicted targets. To make siRNA therapeutics feasible, numerous efforts have been made to reduce their off-target effects, which limit their therapeutic effect [73]. 

Many interesting previously published reports focus on the design of siRNA and miRNA for therapeutic applications [67,74]. One of the most recently approved drugs based on siRNA, inclisiran (LEQVIO; Novartis), is a conjugate of N-acetylgalactosamine (GalNac)-modified siRNA, which is shown to decrease cholesterol levels in plasma. It works via an RNAi mechanism of action and could ameliorate outcomes for patients with atherosclerotic cardiovascular disease. To the best of our knowledge, there are four other drugs currently in the clinical phase 3: (1) vutrisiran (NCT04153149), for the treatment of transthyretin-mediated amyloidosis, including both hereditary and wild-type amyloidosis [75]; (2) tivanisiran (NCT04819269), a topically designed siRNA for the treatment of dry eye disease; (3) teprasiran (NCT03510897), a siRNA used for the temporary inhibition of p53-mediated cell death for the treatment of acute kidney injury; and (4) fitusiran (NCT03417102), a siRNA therapeutic targeting the antithrombin to rebalance hemostasis in people with hemophilia A or B. There are two main therapeutic strategies based on miRNAs: miRNA inhibition therapy and miRNA replacement therapy. Inhibition therapy is similar to the antisense approach, which involves the singles-stranded RNAs that behave as miRNA antagonists for the inhibition of endogenous miRNA. Replacement therapy involves synthetic miRNAs (called mimics) that replace miRNA’s expressions that are normally repressed [76]. 

miRNAs modulate both physiological and pathological processes, playing a crucial role in regulating cell functions such as proliferation, differentiation, and apoptosis. Overexpression or underexpression of miRNAs are associated with many diseases, including cancer. The deregulation of miRNAs could contribute to tumor onset, development, invasion, and metastasis, which are known to be the hallmarks of cancer. 

miRNAs that are overexpressed in cancer are called oncomiR and are the targets of inhibition therapy, whereas miRNAs that are downregulated and that act as tumor suppressors are targets of so-called replacement therapy (Figure 4) [77].

In miRNA replacement therapy, the tumor suppressor miRNAs are replaced by synthetic miRNAs to restore lost function resulting from the downregulation of key miRNAs, for example, let-7 and miRNA-34a to achieve the same biological function as endogenous miRNAs. 

The double-stranded mimic is made of a guide strand (the one actively recognizing the target) that has a similar sequence in its nucleotides, and a passenger strand, which is complementary to the guide, that is discharged during the process. The double structure can facilitate the loading of RNA molecules into the RISC, thereby promoting gene silencing. In vivo, miRNA therapeutics could activate the innate immune system through Toll-like receptors [78] and could generate off-target effects leading to significant undesirable effects. Chemical modification of natural oligonucleotides is the major approach to ameliorate stability and cellular uptake. The appropriate dose of miRNA therapeutics is crucial for boosting their efficiency and potency in cell/tissue and preventing the immunogenic response [8]. The first synthetic potential miRNA to enter clinical trials was MRX34 (NCT01829971) for cancer treatment [79]. For primary liver cancer or liver metastasis from other solid tumors, MRX34 was formulated to deliver a mimic of miRNA-34 using liposomes [80]. 

TargomiRs (NCT02369198) consist of a miRNA-16 mimic, an uptake system based on nonliving bacterial minicell nanoparticles, and an antiepidermal growth factor receptor antibody as the targeting moiety. Introducing synthetic exogenous miRNA-16, another tumor suppressor, mimics promoted the inhibition of tumor-promoting gene transcription and therefore tumor growth [81]. Many tumor suppressor miRNAs, such as miRNA-7, miRNA-126, miRNA-143/145, miRNA-200, miRNA-355, and the members of the let-7 families [82,83], have been identified as downregulating oncogenes. The use of double-stranded miRNA mimics has some drawbacks for drug development, for example, the synthesis of the passenger strand doubles the time and cost requirements for its production, and the passenger could cause off-target effects [84]. The use of single-stranded, chemically modified miRNA mimics such as miR-34a, miR-124, miR-122, and miR-216b was also investigated and showed promising effects [85]. A single-stranded miR-126b mimic, with modified UNA, was conjugated with two palmityl chains on the surface of 1-palmitoyl-2-oleoyl-sn-glycero-3-phosphocholine liposomes and was functionalized with the TAT cell-penetrating peptide. The sequence was tested to inhibit *KRAS* in two pancreatic ductal adenocarcinoma cell lines, showing that the NP functionalized with TAT-Pal and 126b-Pal, respectively, induced an ~70% decrease in protein KRAS and ~40% inhibition of colony formation [86]. In their study, Piacenti and colleagues [87] developed a series of single-stranded PNA analogues of miRNA-34a with different lengths to target 3′ UTR mRNA of *MYCN*. miRNa-34a acted as a tumor suppressor in many tumors including neuroblastoma. miRNA-34a is a direct regulator of the *MYCN* oncogene, whose overexpression is a prominent biomarker for neuroblastoma phenotype [88]. 

The PNA/RNA duplexes held very promising features of affinity and stability and in terms of cellular uptake despite the presence of multiple mismatches. Furthermore, an 8-mer PNA sequence, conjugated with a peptide carrier, was found to show moderate internalization in NB Kelly cells without any transfection agents [89].

In another study, Dhuri et al. proposed a tail camp γPNA (tcγPNA-155) to target the oncomiR-155 by base-pairing W-C and Hoogstein to create a stable clamp and inhibit their activity. The sequence γPNA-155 was delivered into tumors and inhibited the growth of the tumor in lymphoma (DLBCL) [90]. Instead of targeting the full length oncomiR, short-cationic PNAs targeting the seed region were proposed as a next-generation antimiR agent. The cation seed-based PNA oligomers were synthesized with three lysine or arginine units, and they were encapsulated in poly(lactic-co-glycolic acid) PLGA NPs to enhance the delivery. Arginine containing antiseed PNA-155 was reported to cause superior inhibition of the target, and systemic delivery of PNA-155 loaded PLGA NPs prevented the growth of tumors (DLBCL xenografts) [91]. The same authors also developed positively charged PLGA/poly-L-histidine–based nanoparticles to encapsulate the full-length PNA-155 targeting the miR-155 and showed a sixfold reduction in the growth of DLBCL xenografts [92].

Additionally, a sequence γPNA (8 mer) targeting the miR-155 seed region was conjugated at the N-terminus with lauric acid (C12) and conferred an amphiphilic shape resulting in self-assembly. The sequence γPNA-155-LA was formulated in vesicles (∼100 nm) with ethanol and demonstrated cellular uptake and the inhibition in vitro of miRNA-155 [93]. Another approach is to target simultaneously multiple oncomiRs to enhance their anticancer activity. 

The PNA sequence, formulated with PLGA NPs and designed to be complementary to oncomiRs-155 and -21, demonstrated an approximately 80% reduction in the viability of lymphoma cells. This exceeded the efficacy of individual treatments, which achieved around a 50% reduction [94].

Both PS and PNA ASOs, when delivered via PLGA NPs, exhibited similar inhibition of miR-155 (~90%) and miR-21 (~50%) in lymphoma cells. Many oncomiRs, including miR-21, miR-10b, and miR 221, were assessed towards blocking the progression of glioblastoma multiforme (GBM), an aggressive tumor. Various synthetic oncomiRs, including miR-10b, miR-21, miR-221, and miR-93, were tested to evaluate their therapeutic activity in GBM. Concomitant use of oncomiRs-155 and oncomiRs-221 integrated with temozolomide was demonstrated to induce apoptosis in the T98G glioma cell line [95].

Targeting specific oncomiRs-10b or -21 has demonstrated anticancer efficacy in vivo models of GBM. The concurrent inhibition of both oncomiRs-10b and -21 in GBM has been evaluated by few studies, though. Wang et al. synthesized two tiny (8-mer) cationic γ-modified PNAs to target the seed region, respectively, of oncomiRs-10b and -21. The oligomers based on modified γPNAs were entrapped in NPs formed by PLA/HPG-CHO and were tested in a mouse model of GBM [96]. 

## 3. Oligonucleotides as Biomimetic Models for Biological Studies

The relevance of ODNs as biomimetic models stems from their ability to replicate and mimic key features of nucleic acids, providing valuable insights into biological processes, gene function, enzymatic activity, molecular recognition, and self-assembly. Their versatility and programmability make them essential tools for advancing our understanding of biology and developing innovative applications in various fields. In particular, researchers can study the binding of proteins, enzymes, or other molecules to specific DNA or RNA sequences, as well as the effects of mutations or damage on the structure and function of DNA or RNA and their repair. For example, modified ODNs containing single lesions, such as uracil or oxidized guanine, have been used for the biochemical characterization of the base excision repair (BER) pathway. The lesion of interest was located on a double-stranded linear oligonucleotide [97], but the construction of DNA repair substrate with a single lesion at a defined site on the circular DNA allowed the fine mapping of the repair patches, thereby distinguishing between the short- and long-patch BER pathways [98,99]. Here, we will delve deeper into the studies conducted on biomimetic models designed to investigate oxidative DNA damage, focusing specifically on the study of cPu [34,35,100]. These tandem-type lesions, observed as modifications among DNA purines, have been detected in the DNA of mammalian cells in vivo [34,100] and are known to have cytotoxic and mutagenic properties [101,102]. They are exclusively formed when hydroxyl radicals attack 2′-deoxyribose units, leading to the generation of C5′ radicals, which subsequently undergo cyclization with the C8 position of the purine base. These lesions can occur in two distinct diastereomeric forms, namely, 5′*R* and 5′*S* (Figure 5) [100,103,104,105,106].

Understanding the formation, repair, and biological consequences of cPu lesions is crucial for gaining insights into the mechanisms of DNA damage and repair as well as their implications in human health and disease [107]. Studies conducted on single nucleosides and 21-mer model double-stranded ODNs, 5′-d[(GGGTTA)_3_GGG]-3′, allowed the highlighting of the mechanism underlying the formation of these lesions [36,108]. By employing dispersion-corrected density functional theory calculations, tailored computational investigations were conducted, focusing on the double-stranded models of CGC and TAT for each diastereoisomer of cPu. The examination of complete reaction pathways offers valuable molecular explanations for the observed diastereoisomeric selectivity during cPu formation [36]. The presence of oxygen was found to significantly affect the formation of cPu lesions. This observation suggests an alternative reaction pathway for the C5′ radical, where it reacts with oxygen instead of undergoing intramolecular cyclization. Experimental evidence revealed the formation of hydrated 5′-aldehydes rather than cPu when dG and dA are formed in the presence of 2.7 × 10^−4^ M of O_2_, indicating that the C5′ radicals react with molecular oxygen, inhibiting cyclization [109]. 

To further explore the impact of oxygen on the cPu formation, the levels of the four distinct cPu lesions per 10^6^ nucleosides (dA or dG) per Gy of irradiation were assessed in 21-mer ds-ODNs under varying oxygen concentrations. Notably, higher oxygen concentrations were found to decrease the amounts of cdA and cdG, providing experimental confirmation of competing pathways for the C5′ radical between cyclization and addition to oxygen. Interestingly, the 5′*R*/5′*S* diastereomeric ratio for both cdG and cdA remained unaffected by the presence of oxygen, indicating that oxygen does not influence the stereochemistry of cPu formation.

The reaction of hydroxyl radicals (HO•) generated through Fenton reactions with 21-mer ds-ODNs was examined. Furthermore, similar experiments were conducted using ionizing radiation or Fenton reactions with calf thymus DNA, helping to clarify the ongoing debate in the scientific literature regarding absolute levels of cPu lesions, methods of HO• radical generation, 5′*R*/5′*S* diastereomeric ratio, and relative abundance of cPu compared to 8-oxo-Pu [108,110]. A recent study conducted on wild-type and defective Cockayne cell cultures highlighted the absence of cPu lesions in mitochondrial DNA compared to nuclear DNA, suggesting the absence of reactivity of HO• radicals within the mitochondria [111].

The formation of purine lesions also appears to be influenced by the conformation of the DNA. A study carried out by Chatgilialoglu et al. highlighted how folded G-quadruplex sequences exhibit significantly higher susceptibility to purine hydroxyl radical oxidation compared to their unfolded states, underlining the greater reactivity of guanine residues within G-quadruplex structures [112]. 

To conduct epidemiological investigations and comparative studies on the presence of these lesions in living organisms and evaluate the effectiveness of known enzymatic repair systems, it is crucial to establish robust synthetic and analytical methodologies. The main challenge lies in the synthesis of the four distinct cPus, involving numerous synthetic steps, low yields, and the difficult separation of *R* and *S* diastereoisomers. The growing interest in studying these lesions has led to the search for a shorter and higher-yielding synthetic pathway to obtain all four cPus and the related phosphoramidites necessary for the automated construction of model ODNs. Additionally, to recognize DNA damage, a comprehensive molecular library approach was developed, including both labeled and unlabeled compounds, to produce the required analytical standards [113,114,115]. Studies performed on oligonucleotide models incorporating each of the four lesions revealed that the presence of cPu causes a significant structural distortion in the DNA double helix, leading to local conformational changes. This destabilization affects the efficiency of purine lesion repair, with the 5′*R* diastereomer being repaired more efficiently than the 5′*S* diastereomer [37,116,117]. The impact of lesions on the thermodynamic stability of DNA duplexes was investigated by several groups, revealing decreased melting points of the modified duplex sequences compared to their unmodified counterparts. 

A series of sequences, both modified and unmodified, of various lengths and compositions, were synthesized to measure the different melting temperatures and demonstrate the instability induced by the presence of the lesions (Table 1). Furthermore, it was observed that shorter DNA duplexes are more susceptible to destabilization caused by lesions than longer duplexes.

The quantification of these lesions is a crucial step for both mechanistic and biological studies. Significant efforts have been made by various research groups to accurately determine the quantity of these lesions in DNA through enzymatic digestion followed by chromatographic methods coupled with mass spectrometric analysis. Enzymatic digestion is a fundamental step to ensure the complete hydrolysis from the DNA backbone of all four cyclonucleoside lesions. Initially, enzymatic digestion protocols were developed based on the hydrolysis efficiency of modified ODNs containing the lesions [114,115,119].

The protocols developed using ODN substrates were tested, and a new one was developed based on the enzymatic combination of benzonase and nuclease P1 on specifically oxidized calf thymus DNA [119], which better mimics real samples. This method, which involves the use of the enzymatic combination followed by analysis through liquid chromatography coupled with tandem mass spectrometry with isotopic dilution, offers advantages in terms of cost and efficiency for measuring DNA damage, including cyclopurine lesions [120]. It is well established that cPus are exclusively repaired through the nucleotide excision repair (NER) process, although with low efficiency. Studies conducted on model oligonucleotides, each containing a cPu lesion, demonstrated that the 5′*R* diastereomer is more effectively repaired than the 5′*S* diastereomer for both 5′,8-cyclo-2′-deoxyadenosine (cdA) and 5′,8-cyclo-2′-deoxyguanosine (cdG) [37]. These findings highlight the biological importance of the stereochemistry of cPus. Furthermore, both *R* and *S* stereoisomers demonstrate weakly destabilizing effects within the nucleosome, preventing rapid access to the NER pathway. The observed resistance to NER in vivo suggests a reduced susceptibility to efficient repair mechanisms [121].

If not repaired, these mutations can contribute to genomic instability and potentially lead to the development of various diseases, including cancer and neurodegenerative diseases [34,122,123,124,125,126,127]. Recent studies revealed that both cdA and cdG robustly block transcription and are efficiently repaired by NER. Notably, the components of transcription-coupled (TC)-NER, specifically CSB/ERCC6 and CSA/ERCC8, were found to be as essential as XPA. These observations underscore the strict requirement of TC-NER for the efficient removal of cdA and cdG lesions, highlighting them as potential contributors to the cytotoxic and degenerative responses observed in individuals with genetic defects in this pathway. In this research, the synthetic 18-mer oligonucleotides 5′-d(CATTGCTTCGCT [5′*S*-cdA]GCACG)-3′ and 5′-d(CATTGCTTC [5′*S*-cdG]CTAGCACG)-3′ were integrated into an enhanced green fluorescent protein reporter gene for assessing the inhibitory effects of these alterations on transcription in human cells [128].

Although no glycosylases have been identified to recognize cdA and cdG structures, rendering them unsuitable substrates for BER, several studies demonstrated that their presence can significantly impact the efficiency of nuclear BER. In particular, Karwowski and colleagues investigated the influence of 5’*R* and 5’*S*-cdA on the BER machinery using chinese hamster ovary, both nuclear and mitochondrial, cell extracts. These investigations involved the utilization of synthetic oligonucleotides containing double-stranded clustered DNA lesions (CDLs). These oligonucleotides consisted of one strand harboring one of the four diastereomeric lesions, while the other strand contained an apurine/apyrimidine site (AP site) positioned at varying interlesion distances. The outcomes of these studies demonstrate that the presence of 5′*R*-cdA exhibits a more pronounced inhibitory effect on BER compared to 5′*S*-cdA, whereas 5′*S*-cdG leads to a more significant reduction in the repair level than 5′*R*-cdG [129,130,131].

Furthermore, the results obtained from the bacterial model, *Escherichia coli* reporter assay, together with the use of 40-mer model ds-oligonucleotides containing CDL, align with previous in vitro investigations conducted on eukaryotic models. The observed high mutagenicity and/or impeded repair process in clusters where the cdA lesions are located in close proximity provide additional confirmation of the previously proposed patterns that describe the impact of the distance between cdA and dU on DNA repair processes [132]. Several investigations were conducted on pol β, utilizing substrates containing a 31-nucleotide template strand with either a 5′*R*-cdA or 5′*S*-cdA positioned at the 19th nucleotide from the 3′-end. These substrates were designed to simulate DNA replication or BER processes. The studies revealed that when 5′*R*-cdA is bypassed by pol β, it can result in nucleotide misincorporation, leading to mutations. Conversely, the bypass of a 5′*S*-cdA by pol β can lead to the accumulation of DNA strand break intermediates, causing recombination events and genome instability [133]. Interestingly, although pol β experiences a stall when encountering a 5′*S*-cdA within random sequences, it efficiently extends a dT nucleotide opposite to a 5′*S*-cdA located within trinucleotide repeats, such as CAG repeats. This phenomenon leads to CTG repeat deletion through the BER pathway. The underlying reason for this lies in the fact that both the *R* and *S* diastereomers of cdA located on the template strand cause the formation of a CAG repeat loop, mimicking the intermediates formed during the maturation of the lagging strand and BER [134].

These findings provide further evidence of the crucial role played by pol β in bypassing loop structures containing cPu lesions, ultimately mediating the deletion of trinucleotide repeats. Poly (ADP-ribose) polymerase 1 (PARP1), an ADP-ribosylating enzyme found also to be involved in different DNA repair mechanisms, is able to selectively recognize and distinguish between two types of damaged DNA lesions, specifically 5′*S*-cdA and 5′*R*-cdA. Interestingly, PARP1 has exhibited different binding affinities for double-stranded DNA substrate (23-mer) containing these lesions, with a higher affinity for 5′*S*-cdA compared to 5′*R*-cdA, indicating its ability to differentiate between the diastereomeric forms of the lesions [118].

## 4. Conclusions

In conclusion, this review provided insights into future trends and challenges in nucleic acid research, highlighting the enormous potential and impact of this field. Developments in these areas will not only deepen our understanding of fundamental biological processes but also pave the way for new diagnostic tools, personalized medicine, and innovative nanodevices. The importance of designing ASOs with high specificity, allowing for precise targeting of individual genetic differences or the accurate identification of disease-related biomarkers, will enable a personalized approach to treatment. Therapies can be tailored to meet the specific needs of each patient, potentially leading to more effective and efficient medical interventions. Overcoming challenges related to nucleic acid delivery, improving the safety and efficacy of therapies, and advancing nucleic acid-based technologies will be crucial to realizing the full potential of nucleic acids in medicine, biotechnology, and beyond. However, through collaborative efforts, interdisciplinary approaches, and ongoing technological advancements, the future of nucleic acid research appears promising, offering immense opportunities for scientific progress and the enhancement of human health.

## Figures and Tables

**Figure 1 ijms-25-00146-f001:**
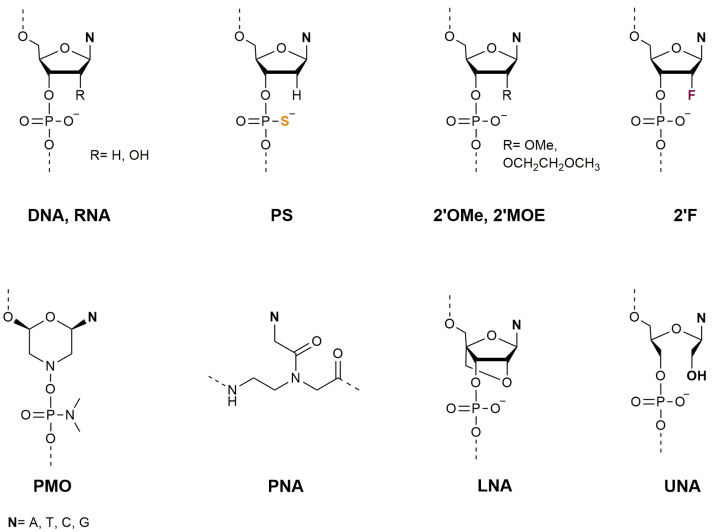
Examples of nucleic acid analogues featuring modifications to the backbone and sugar. PS: phosphorothioate; 2′OMe: 2′-O-methyl; 2′MOE: 2′-O-methoxyethyl, 2′F: 2′-fluoro; PMO: mor-pholino phosphorodiamidate oligomer; PNA: peptide nucleic acid; LNA: locked nucleic acid; UNA: unlocked nucleic acid.

**Figure 2 ijms-25-00146-f002:**
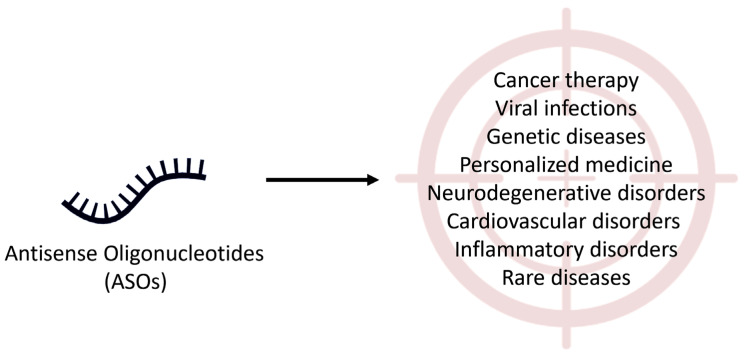
Indicative areas of application of antisense drugs. This figure was created by using Bio-render (www.biorender.com, accessed on 1 November 2023).

**Figure 3 ijms-25-00146-f003:**
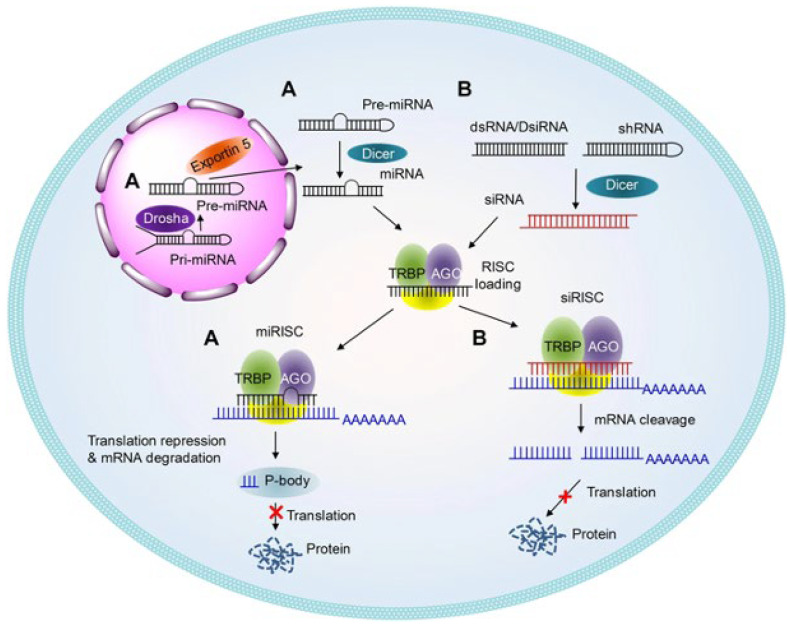
Schematic illustrations of the working mechanisms of miRNA (A) and siRNA (B). The precursors of siRNA and miRNA are produced in the nucleus and are substrates of Drosha, which produces a short hairpin framework. Then, these structures are exported by Exportin-5. Another enzyme, called Dicer, processes the short hairpin, producing siRNAs and miRNAs. The siRNA/miRNA are incorporated into RISC complex. One of the strands, known as the “guide” strand, binds to the target mRNA within the RISC complex. Perfectly complementary binding results in the degradation of the target mRNA, while partial complementarity leads to inhibition of protein synthesis. Reprinted with permission from Ref. [9].

**Figure 4 ijms-25-00146-f004:**
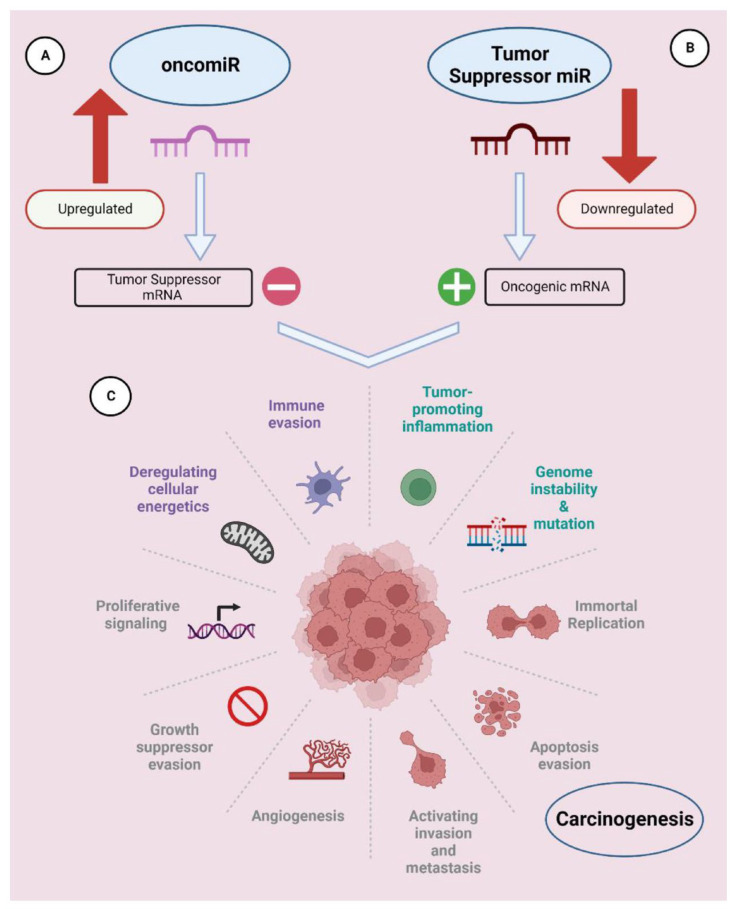
The regulatory role of oncogenic and tumor suppressor microRNAs in tumorigenic events. (**A**) The constitutive overexpression of oncomiR hinders the translation of tumor-suppressor genes, fostering the growth of tumor cells; (**B**) Through the suppression of mRNA translation encoding oncogenes, tumor-suppressor miRNAs impede tumorigenesis and the ensuing development of cancer; (**C**) Key characteristics of the process leading to the formation of cancer. Reprinted with permission from Ref. [77].

**Figure 5 ijms-25-00146-f005:**
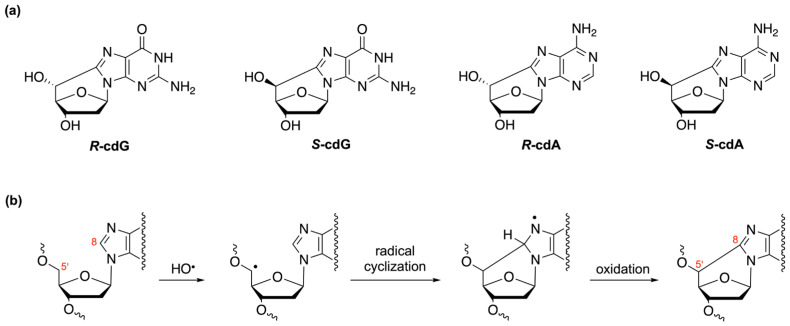
(**a**) Structures of 5′,8-cyclo-2′-deoxyadenosine (cdA) and 5′,8-cyclo-2′-deoxyguanosine (cdG) in their 5′*R* and 5′*S* diastereomeric forms. (**b**) Purine 2′-deoxynucleoside moiety reacts with hydroxyl radical (HO•) yielding the cPu via C5′ radical cyclization followed by oxidation.

**Table 1 ijms-25-00146-t001:** The melting point (Tm) measurements of a series of oligonucleotide sequences containing the diastereoisomers of 5′,8-cyclopurine lesions. Adapted with permission from Ref. [35].

Oligomer Lengths	Sequences (Double-Stranded)	Melting Temperatures (Tm °C)	References
11	5′-d(CGT ACX CAT GC)-3′ 3′-d(GCA TGT GTA CG)-5′	49.5 (X = dA) 42.0 (X = 5′*S*-cdA)	[116]
12	5′-d(GTG CXT GTT TGT)-3′ 3′-d(CAC GCA CAA ACA)-5′	55.0 (X = dG) 46.0 ± 1 (X =5′*S*-cdG)	[117]
14	5′-d(ATC GTG XCT GAT CT)-3′ 3′-d(TAG CAC TGA CTA GA)-5′	54.0 ± 1 (X = dA) 48.0 ± 1 (X = 5′*S*-cdA)	[109]
17	5′-d(CCA CCA ACX CTA CCA CC)-3′ 3′-d(GGT GGT TGT GAT GGT GG)-5′	65.2 ± 0.6 (X = dA) 58.9 ± 0.6 (X = 5′*R*-cdA) 60.5 ± 0.6 (X = 5′*S*-cdA)	[37]
17	5′-d(CCA CCA ACX CTA CCA CC)-3′ 3′-d(GGT GGT TGC GAT GGT GG)-5′	66.2 ± 0.7 (X = dG) 63.4 ± 1.0 (X = 5′*R*-cdG) 63.5 ± 0.6 (X = 5′*S*-cdG)	[37]
23	5′-d(GCA GAC ATA TCC TAG AGX CAT AT)-3′ 3′-d(CGT CTG TAT AGG ATC TCT GTA TA)-5′	60.0 ± 0.3 (X = dA) 59.0 ± 0.2 (X = 5′*R*-cdA) 58.0 ± 0.3 (X = 5′*S*-cdA)	[118]

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
