# Peer review of "Advances in Nucleic Acid Research: Exploring the Potential of Oligonucleotides for Therapeutic Applications and Biological Studies"

_ijms, 2023, doi:10.3390/ijms25010146_

Round 1
Reviewer 1 Report
Comments and Suggestions for Authors
Minor comments:
(1) Overall, quality of figures should be enhanced. Captions should not use indentation.
(2) There is no description concerning PMO and UNA in Figure 1.
(3) Typo in Figure 2 caption: neurodegenerative disorders
(4) Figure 3: unnecessary outline, the mRNA figure is bad.
(5) Figure 4: resolution is not acceptable.
(6) Table 1: separated caption
Major comments:
(1) I suggest inserting a table of figure to summarize the stuffs described, so that readers can find information at a glance.
Although the theme and text within the manuscript cover timely topics, the quality of the figures are low. Thus, the authors should revise the figures thoroughly. Comments on the Quality of English LanguageN/A
Reviewer 2 Report
Comments and Suggestions for Authors
In the Review Article entitled "Advances in Nucleic Acid Research: Exploring the Potential of Oligonucleotides for Therapeutic Applications and Biological Studies" by Moccia M. et al., the authors reviewed the recent literature on the exploitation of the fast-growing toolbox of synthetic natural and modified nucleic acids analogs in therapeutics and biotechnological applications. The topic is of the utmost importance, considering the raising interest towards the development of the so-called "personalized medicine" aimed both at reducing the off-target effects of conventional therapeutics and enhancing the therapeutic efficacy towards genetic diseases and cancer. In addition, I found very interesting the paragraph dedicated to the biomimetic models designed to investigate the DNA oxidative damage caused by the formation of cyclo-purines.
The submitted review is well organized and easy to follow. However, in its present form, it is not suitable for publication because of the presence of many English Language issues (some of which are listed in the Comments on the Quality of English Language box below) that have a detrimental effect on the readability of the manuscript.
The reference list is comprehensive and focused on very recent literature.
The authors clearly stated that their review is focused on the antisense and RNA interference strategies. However, I believe that the inclusion of a brief overview on the application of oligonucleotide analogues (PNAs and miRNAs above the other) in the antigene approach would be very useful to allow the readers to get the full picture of the therapeutic potentialities of ONs.
Minor points:
1) the acronym "PMO" should be used instead of "MF" at page 5, line 179, and it should be spelled out as "morpholino phosphorodiamidate oligomer" rather than "morpholino phosphoroamidates".
2) the acronym "MAPT" should be spelled out at page 6, line 252.
3) "2'-deoxy-2-fluoro-D-arabinonucleic acid" should be replaced by "2'-deoxy-2'-fluoro-D-arabinonucleic acid" at page 7, line 299.
4) the acronym "BER" should be spelled out at its first occurrence at page 11, line 467.
5) "cdPu" should be "cPu" at page 15, line 629.
6) remove bold style to the period in "(Table 1)." at page 13, line 20.
7) fix "in in" at page 11, line 450.
8) fix "to to" at page 12, line 523.
9) replace hyphen with minus sign in "2.7 x 10-4" at page 12, line 502.
10) fix the word "Neurodegenerative" in Figure 2 that is splitted in two lines.
Based on the above considerations, I can recommend publication after the above-mentioned suggestions and the English language editing will be fully addressed.
Comments on the Quality of English Language
The manuscript requires a comprehensive revision of the English language as some sentences are not immediately understandable. Special attention should be given to the consistency of tenses, use of singular and plural forms, prepositions, and articles. A non-exhaustive list of necessary corrections is provided below:
1) "a phosphate group" at page 2, line 54
2) "ODNs have been" at page 2, line 85
3) fix "mecha-nism" at page 6, line 202
4) "For example, targeting the oncogene BCL-2 with ASOs found to induce apoptosis in cancer cells and inhibit tumor formation" at page 7, lines 264-265 needs rephrasing
5) fix "they have a primary roles" at page 8, line 315
6) fox "imporant" at page 8, line 322
7) check and fix the tenses accordance in the caption of Figure 4
8) consider replacing "depends by" with "depends on" at page 9, line 340
9) remove comma in "efficacy, [70,71]." at page 9, line 343
10) "Chemical modification of natural oligonucleotides is the major approach to ameliorate the stability and the cellular uptake to solve this problem." at page 10, lines387-388. "To solve this problem" should be removed to avoid possible misunderstanding of the sentence.
11) The sentence "Appropriate dose of miRNA therapeutics for boosting their efficiency and potency in cell/tissue to avoid the immunogenic response [8]." at page 10, lines 389-390, is incomplete (the verb is missing).
12) At page 10, lines 394-396, replace "consists" with "consist"; consider replacing "based on nanoparticle using non-living bacterial minicells" with "based on non-living bacterial minicell nanoparticles" and "as a targeting" with "as the targeting moiety".
13) Replace "have" with "has" at page 10, line 401.
14) The sentence "Piacenti et al. [85] developed a series of single stranded PNA analogues of miRNA-34a, with different length, to target 3’ UTR mRNA of MYCN an oncogene.".
15) Page 10, line 430, replace "author" with "authors".
16) Page 11, line 438, consider replacing "simultaneous" with "simultaneously".
17) The sentence "The PNA sequence, formulated by NPs with PLGA, complementary to oncomiRs-155 and -21 has demonstrated reduction in viability of lymphoma about ~80% cells compared to the single treatments (~50%) [92]." needs rephrasing (page 11, lines 439-441).
18) "Both phosphorothioate (PS and PNA ASOs)" at page 11, line 442, should be "Both phosphorothioate (PS) and PNA ASOs".
19) At page 11, lines 443-444, "Many oncomiR such as miR-21, miR-10b, miR 221 were assessed" should be rewritten as "Many oncomiRs, including miR-21, miR-10b, and miR 221 were assessed".
20) Fix "glioblastoma multiforme (GBM) an aggressive tumors" with "glioblastoma multiforme (GBM), an aggressive tumor" at page 11, line445.
21) Check the sentence at page 11, lines 452-453, "Wang et al., have synthesised a tiny (8-mer) cationic γ-modified PNAs to target the seed region respectively of oncomiRs-10b and -21.". Consider replacing "a tiny" with "two tiny".
22) Related to point 21. "oligomer" should be "oligomers" and "nanoparticle" should be "nanoparticles" at page 11, lines 453-454.
23) Fix the sentence "These tandem-type lesions are observed as modifications among DNA purines, have been detected in mammalian cellular DNA in vivo [32,98],..." at page 11, 473-474.
24) Replace "an 5'S-cdA" with "a 5'S-cdA" at page 15, lines 619, 621, and 622.
Round 2
Reviewer 1 Report
Comments and Suggestions for Authors
Moccia et al. report the manuscript entitled 'Advances in Nucleic Acid Research: Exploring the Potential of Oligonucleotides for Therapeutic Applications and Biological Studies' as a review in IJMS at MDPI.
In the previous review report, I suggested major revision in all figures. I found figure 1 and 2 with no changes. The authors are able to produce high quality chemical structure like figure 5. But the quality of figure 1 is rather lower than the quality of figure 5. I suggest redrawing as I depicted shortly.
https://ibb.co/ckdKmrr
For your information in figure 1: (i) please use sane bond length (ii) no weird bond angle (iii) align all nucleosides in same row evenly (iv) instead of B, use N representing nucleobases (v) highlight bond or atoms or interest (vi) put some caption text like - figure 3 - to give more information to readers.
Figure 2 needs extensive revision as it appears as a freshman's report.
In Figure 3, four bulleted and capitalized hallmark claims are not well-described in main text. In my dictionary I could not find 'METHASTASIS'.
I must reject the current version if the qualities of figures 1, 2 and 4 are not significantly enhanced.
Reviewer 2 Report
Comments and Suggestions for Authors
The authors addressed in full all points and suggestions.
Author Response
We would like to express our gratitude to the reviewer for the valuable suggestions provided, which undoubtedly contributed to improving the manuscript.
Round 3
Reviewer 1 Report
Comments and Suggestions for Authors
The authors addressed concerns that I raised.